Regulating the UAS/GAL4 system in adult Drosophila with Tet-off GAL80 transgenes

Barwell Taylor 1
DeVeale Brian 1 2
Poirier Luc 1
Zheng Jie 1 3
Seroude Frederique 1
Seroude Laurent seroudel@queensu.ca 1
1 Department of Biology, Queen’s University , Kingston , ON , Canada
2 Current affiliation:  Department of Urology, UCSF School of Medicine , San Francisco , CA , United States of America
3 Current affiliation:  Institute for Biomedical Informatics, University of Pennsylvania , Philadelphia , PA , United States of America
Singh Shree Ram
Electronic publication date: 2017 Dec 14
Publication date: 2017
Volume: 5
Electronic Location ID: e4167
Received 2017 Jul 26; Accepted 2017 Nov 24
Copyright: ©2017 Barwell et al.
Copyright year: 2017
Copyright holder: Barwell et al.
License: This is an open access article distributed under the terms of the Creative Commons Attribution License, which permits unrestricted use, distribution, reproduction and adaptation in any medium and for any purpose provided that it is properly attributed. For attribution, the original author(s), title, publication source (PeerJ) and either DOI or URL of the article must be cited.
License URL: https://creativecommons.org/licenses/by/4.0/

Keywords: Aging, Muscle, Oenocytes, Grim, Gene expression, tTA, TetO, Actin-5c, Tubulin

Funding: Institute of Aging of the Canadian Institutes of Health Research MOP64248 MOP79519 Natural Sciences and Engineering Research Council of Canada RGPIN/250140-2010 This work was supported by the Institute of Aging of the Canadian Institutes of Health Research (CIHR, MOP64248, MOP79519) and the Natural Sciences and Engineering Research Council of Canada (RGPIN/250140-2010). The funders had no role in study design, data collection and analysis, decision to publish, or preparation of the manuscript.

==============================
The UAS/GAL4 system is the most used method in Drosophila melanogaster for directing the expression of a gene of interest to a specific tissue. However, the ability to control the temporal activity of GAL4 with this system is very limited. This study constructed and characterized Tet-off GAL80 transgenes designed to allow temporal control of GAL4 activity in aging adult muscles. By placing GAL80 under the control of a Tet-off promoter, GAL4 activity is regulated by the presence or absence of tetracycline in the diet. Almost complete inhibition of the expression of UAS transgenes during the pre-adult stages of the life cycle is obtained by using four copies and two types of Tet-off GAL80 transgenes. Upon treatment of newly emerged adults with tetracycline, induction of GAL4 activity is observed but the level of induction is influenced by the concentration of the inducer, the age, the sex and the anatomical location of the expression. The inhibition of GAL4 activity and the maintenance of induced expression are altered in old animals. This study reveals that the repressive ability of GAL80 is affected by the age and sex of the animal which is a major limitation to regulate gene expression with GAL80 in aged Drosophila.

Introduction

The ability to make targeted gene manipulations is crucial to the investigation of biological phenomena and is the hallmark of modern genetics. Inducible gene expression systems, which allow for the temporal and spatial regulation of expression of a gene of interest, have been successfully employed in identifying and studying genes that influence aging in the model organism Drosophila melanogaster (Poirier & Seroude, 2005). Various inducible-gene expression systems are available in Drosophila, the majority of which require the use of two distinct transgenic constructs containing respectively an effector (EG) and a transactivator (TA) gene (Venken, Simpson & Bellen, 2011). The expression of the effector gene is controlled by a promoter, which is regulated by the transactivator. Thus, the expression pattern of the EG is indirectly controlled by the nature of the promoter directing TA expression. The most widely used expression system is the UAS/GAL4 system in which the EG uses a promoter containing upstream activating sequences (UAS) and the TA is the yeast transcription factor GAL4 that can be controlled by an enhancer-trap or a Drosophila promoter (Brand & Perrimon, 1993). Since its introduction, thousands of GAL4 lines (GAL4 drivers) have been generated. The majority of the GAL4 drivers available are solely useable during the pre-adult stages because the expression pattern of GAL4 across the life span of the adult is generally ignored, despite the fact that both the magnitude and localization of expression can change with age (Seroude et al., 2002). However, many have been characterized and are suitable to manipulate gene expression in aging adults (Seroude, 2002; Seroude et al., 2002; Shen et al., 2009).

Because most promoters and enhancers that drive GAL4 expression are active at multiple stages of development, temporal control of transgene expression is impossible with the UAS/GAL4 system. Temporal control of the UAS/GAL4 system has been achieved by replacing GAL4 with the Gene-Switch protein, which is a chimeric GAL4 protein that contains the GAL4 DNA binding domain, the human progesterone receptor ligand-binding domain and the activation domain from the human protein p65 (Osterwalder et al., 2001; Roman et al., 2001). Only in the presence of the antiprogestin RU486 does the chimeric molecule bind to the UAS sequence and activate transcription. The main restraint with the Gene-Switch system is that since it uses a chimeric form of GAL4, driver lines need to be generated and characterized. Additionally, for most major tissue types in Drosophila, including muscle, fat and nervous tissue, there exist no Gene-Switch drivers that are suitable for aging studies (Poirier et al., 2008). The limitations of the available drivers typically arise from the fact that uninduced transcriptional activity and expression outside of the intended target tissues is observed.

Inducible gene expression systems providing the ability to control the timing of expression have been developed such as the tetracycline-dependent systems that rely on chimeric TAs, either the tetracycline-responsive transactivator (tTA) or the reverse tetracycline transactivator (rtTA) (Gossen & Bujard, 1992; Gossen et al., 1995). Both are based on a fusion between the tetracycline repressor protein from E. coli and the activation domain of the VP16 protein from Herpes Simplex Virus. The gene of interest is placed under the control of a promoter containing tetracycline operator sequences (TetO), to which the transactivators bind to regulate the activity of the promoter. When the tTA transactivator is used, the system is referred to as Tet-Off because tTA only binds the operator sequences in the absence of tetracycline. Conversely, rtTA only binds the operator sequences in the presence of tetracycline, and the system is called Tet-On. Thus, in each system the expression can be regulated temporally by excluding or including tetracycline, or its derivative doxycycline, in the diet (Bello, Resendez-Perez & Gehring, 1998; Bieschke, Wheeler & Tower, 1998; Stebbins et al., 2001). It is worth noting that doxycycline inducing levels do not affect the longevity of experimental animals (Bieschke, Wheeler & Tower, 1998). By starting the feeding after adult emergence developmental effects resulting from pre-adult transgene expression can be avoided, making this system better suited for aging studies. The major limitation is that it requires the generation and characterization of TA driver lines, as well as building TetO transgenic lines. Furthermore, the induction is not observed systematically and requires screening for TA/EG combinations that results in robust induction levels (Bieschke, Wheeler & Tower, 1998). Additionally, in the absence of the inducer the reporters tested in the literature are leaky when the activator is present (Bieschke, Wheeler & Tower, 1998; Stebbins et al., 2001). To allow for greater control over spatial expression and to avoid having to generate and characterize TA lines, the Tet system has been combined with the UAS/GAL4 system (Stebbins et al., 2001; Stebbins & Yin, 2001). A standard GAL4 driver controls expression of either UAS-tTA or UAS-rtTA transgenes. A third transgene is needed in which the EG is controlled by a TetO operator. However, this system has only been examined in the embryonic central nervous system, larval third-instar wing disc, and adult head.

The most recently established system offering temporal control is the Q-system that exploits the two regulatory genes from the Neurospora crassa qa gene cluster. The first transgene contains the TA, QF, that binds the upstream activation sequence, QUAS, within the second transgene driving expression of the EG (Potter & Luo, 2011; Potter et al., 2010; Riabinina et al., 2015). An additional construct contains the repressor, QS, that binds to QF and prevents gene expression. This repression can be relieved by the presence of quinic acid in the diet. The Q system was initially reported to cause detrimental effects due to QF toxicity, which have now been greatly reduced through protein engineering (Riabinina et al., 2015). However, the effect of the inducer on longevity has not been examined.

Figure 1 (A) Temporal control of the UAS/GAL4 system by Tet-off GAL80 transgenes. (B) Two different Tet-off GAL80 plasmids were generated with different promoters driving tTA expression: pDJ146 (tubulin 1α) and pDJ147 (actin5C). Block arrows, promoters; Pentagon, endogenous enhancers; Lozenges, P-element inverted repeats; Triangles, insulator sequences; Ovals, proteins; Rectangles, coding sequences or exons; Rounded rectangles, transcription termination sequences; Striped rectangle, pUC8 plasmid sequence.

Although promising, the systems that allow for temporal control are still relatively new and thus do not yet have as many TA and EG lines as the UAS/GAL4 system. It is a great benefit to fly geneticists to have as many distinct systems as possible in order to be able to target multiple genes simultaneously or achieve intersectional targeting (Dolan et al., 2017; Pfeiffer et al., 2010; Potter & Luo, 2011; Potter et al., 2010; Riabinina et al., 2015; Venken, Simpson & Bellen, 2011). It would be convenient to design a system that makes use of components that have been previously built and characterized and therefore takes advantage of the existing components of the UAS/GAL4 system. In this report the Tet-Off system is used to create “Tet-off GAL80” transgenes that confer temporal control of the UAS/GAL4 system in aging adults while preserving the ability to use the multitude of UAS and GAL4 lines available (Fig. 1A). These transgenes relies on the GAL80 protein to regulate GAL4 transcriptional activity. The GAL80 protein antagonizes GAL4 activity by binding to the activation domain of GAL4, preventing interaction between GAL4 and the transcriptional machinery (Ma & Ptashne, 1987; Nogi & Fukasawa, 1989; Wu, Reece & Ptashne, 1996; Yun et al., 1991). The ability of GAL80 to repress GAL4 transcriptional activity in flies has been demonstrated numerous times with GAL80 being expressed under Drosophila promoters (Lai & Lee, 2006; Lee & Luo, 1999), enhancer traps (Suster et al., 2004), and the TARGET system (McGuire et al., 2003). However, there are a limited number of Drosophila promoters that have been identified and characterized, and by expressing GAL80 through enhancer traps or promoters, the system still lacks temporal control. Although the TARGET system does allow for temporal control, the reliance on heat shock to induce the system prohibits its use in most adult and aging studies, since longevity, behavior and physiology are sensitive to temperature (Ashburner, Golic & Hawley, 2005). By regulating GAL80 expression with the “Tet-Off” tetracycline-dependent inducible gene expression system, GAL80 expression can be controlled. In the absence of inducer, GAL80 inhibits GAL4 activity, thereby preventing the expression of the UAS transgene. Upon addition of inducer, GAL80 expression is switched off, relieving GAL4 of any inhibition. GAL4 then activates the gene of interest that is linked to the UAS sequence. The goal of this study is to use Tet-off GAL80 transgenes to manipulate gene expression in aging adults. Therefore, the absolute priority is to ensure that the expression of a UAS transgene is completely repressed in the absence of the inducer. The GAL4 driver, DJ694, is chosen to test the Tet-off GAL80 transgenes as it is the driver with the highest level of GAL4 expression among the 180 GAL4 lines for which the spatio-temporal expression patterns have been characterized across the lifespan (Seroude et al., 2002). This driver has the added advantage to be expressed in muscle. Sarcopenia, the weakening and loss of skeletal muscle, is one of the most obvious phenotypes associated with aging that drastically affects quality of life (Demontis et al., 2013). Indeed, the elucidation of the molecular mechanism underlying sarcopenia is one of the top priorities of aging research and Drosophila molecular genetics is being increasingly used to tackle this issue (Demontis et al., 2014; Demontis & Perrimon, 2010; Martin et al., 2009; Zheng et al., 2005). This task will be greatly facilitated by the ability to control gene expression in aging muscles. Although a Gene-Switch muscle driver is available (Osterwalder et al., 2001), its activity is insensitive to the presence of RU486 and is therefore useless to achieve temporal control of gene expression in adult muscles (Poirier et al., 2008).

Material and Methods

DNA cloning

Cloning was done using standard molecular biology procedures (Sambrook & Russell, 2001). Escherichia coli XL1Blue (Stratagene, San Diego, CA, USA) cells were transformed by electroporation (Bio-Rad Laboratories, Hercules, CA, USA). Plasmid DNA was purified with the QIAprep or Maxiprep kits (Qiagen, Valencia, CA, USA). DNA fragments were purified with the QIAquick or QIAEX II kits (Qiagen, Valencia, CA, USA). Restriction enzymes, T4 DNA polymerase, T4 DNA ligase, S1 nuclease were purchased from New England Biolabs. PCR were performed with the proofreading PfuTurbo DNA polymerase (Stratagene, San Diego, CA, USA). Oligonucleotide synthesis and DNA sequencing were carried out by Eurofins MWG Operon. The construction and maps of the plasmids (Figs. S2–S12) generated during this study are provided in the supplemental material.

Fly strains

The w[*]; P{w[+mC]=UAS-lacZ.B}Bg4-2-4b (#1777), y[1],w[*]; P{w[+mC]=UAS-mCD8::GFP.L}LL5 (#5137) and w[*]; P{w[+mW.hs]=GawB}how[24B] (#1767) strains were obtained from the Bloomington Stock Center at Indiana University. The UAS-Grim (w1118; P{w[+mC]=UAS-grim-2})(obtained from J Abrams), w[1118]; P{7T40.B}E1(2)[tetO-lacZ] (obtained from J Tower), MHC Geneswitch (obtained from S Pletcher) and w[1118]; P{w[+mW.hs]=GawB}EDTP[DJ694] (#8176) strains have been previously described (Bieschke, Wheeler & Tower, 1998; Osterwalder et al., 2001; Seroude et al., 2002; Wing et al., 1998).

P-element mediated transformations were carried out without removal of the chorion and desiccation (Robertson et al., 1988). 0.2 µg/µl pDJ146 or pDJ147 and 0.04 µg/µl pπ25.7 (Karess & Rubin, 1984) plasmids in 1 mM sodium phosphate buffer pH 7 were injected in 15–45 min old w1118 embryos immersed in Voltalef Oil. This study only used the transgenic strains with an insertion on the X (DJ147 insertions 3.2 and 4) or 3rd chromosome (pDJ146 insertions DJ1077, DJ1078, DJ1079, DJ1080, DJ1081 and DJ1083; DJ147 insertions 2.1 and 3.3). Recombinant strains with 2 DJ146 transgenes were obtained by eye color selection. Recombinant strains with DJ146 and DJ147 transgenes were obtained by PCR selection. The presence of the DJ146 and DJ147 insertions were detected respectively with the DJ120 (5′-AAGTGAGTATGGTGCCTATCTAAC-3′) and DJ140 (5′-AGAGAGACCAAGTGCCATTAC-3′) primers, and with the DJ120 and DJ125 (5′-ATGCGTCATTTTGGCTCG-3′) primers.

UAS-grim lethality test

Strains homozygous for the GAL4 transgenes with 0, 1 or 2 homozygous GAL80 transgenes (experimental) and the w1118 strain (no driver control) were crossed with homozygous UAS-grim strain with 0, 1 or 2 homozygous GAL80 transgenes. For a given cross, 20–25 males were mated with 40–50 virgin females for 48–72 h before 100 to 200 0–12 h old eggs were individually collected and transferred onto plates (25 eggs per plate, four or eight plates) containing standard fly food supplemented with ampicillin (30 µg/ml) or ampicillin (30 µg/ml) plus tetracycline (50 µg/ml). The addition of ampicillin prevents lethality resulting from occasional bacterial contamination of the plate during the egg transfer process. The plates were then maintained at 25 °C. Hatching rate were assessed 26–30 h after egg transfer by scoring the number of empty eggshells. Pupation and adult emergence rates were determined 5–6 days and 10–12 days after egg transfer respectively. The scoring for each stage was converted to the percentage of animals remaining from the previous stage. The orientation, number and parental genotypes of all crosses are provided in Table S1. Inhibition of GAL4 was determined by comparison with negative controls that are lethal (animals without GAL80). Complete inhibition of GAL4 was assessed by comparison with positive controls that displayed no lethality (animals without GAL4 or UAS-grim). Experimental animals that are not significantly different (p < 0.05) from the negative control indicated complete failure to inhibit the expression of the UAS promoter. Experimental animals that are not significantly different (p < 0.05) from the positive control indicated complete repression of the expression of the UAS promoter.

The insertion effects of the GAL80 transgenes were assessed using t-tests between different insertions within the same category of transgene type, copy number, gender and antibiotic treatment (Table S2). All the insertions that are not significantly different were then averaged and the effect of copy number and transgene type (DJ146 vs DJ147) was determined using t-tests (Table S3).

Food preparation

A 10 mg/ml stock solution of tetracycline in water was made. Food with the desired tetracycline concentration was prepared by diluting the stock solution with water and adding 1ml of the diluted solution onto the surface of standard fly food (0.01% molasses/8.2% cornmeal/3.4% killed yeast/0.94% agar/0.18% benzoic acid/0.66% propionic acid). The vials were then allowed to dry at room temperature for 12 to 36 h depending on the ambient humidity.

Fly cultures

DJ694 and w1118 males were crossed with UAS-lacZ females while DJ694; 3.3 + DJ1077 males were crossed with UAS-lacZ; 3.3 + DJ1077 females. Age-synchronized cohorts of the resulting adult progeny were obtained by emptying cultures and collecting newly emerged flies within 48 h. Approximately 350 males and 350 females from each cross were collected under nitrogen anesthesia. Males and females were maintained separately at 25 °C at a density of 25–30 individuals per vial. Each gender was randomly subdivided into 4 cohorts, each of which was provided food containing a different tetracycline concentration (0, 1, 10, or 100 µg/ml). Fresh food was provided by transferring to new vials twice a week. Individuals were randomly removed from each cohort at 2, 5, 10, 20, 30 and 40 days after collection and processed for the CPRG assay. For the X-GAL staining, flies were removed and processed at 10 days and 30 days.

CPRG assay

Samples were manually homogenized in 100 µl extraction buffer (50 mM Na2HPO4/NaH2 PO4, 1 mM MgCl2, pH 7.2, 1 cOmplete™ protease inhibitors cocktail tablet(Roche)/40 ml). L1 samples contained four larvae, while L3, early and late pupae, and adult samples contained one individual per sample. Five independent pre-adult samples and at least three adult samples were tested. Extracts were centrifuged for 1 min at 13,000 rpm. 10 µl of extract supernatant was mixed with 100 µl 1 mM chlorophenolred-ß-D-galactopyranoside (CPRG, Roche) except for L1 samples where 60 µl supernatant was mixed with 50 µl 2mM CPRG. Background levels from the negative controls were averaged and subtracted from the experimental samples. Enzymatic activity was calculated as the rate of change of OD562nm per minute, and standardized to a single fly equivalent (adult samples) or to µg protein content (pre-adult samples). A Bradford Assay (Bio-Rad, Hercules, CA, USA) was used to determine the protein concentrations of the pre-adult samples.

X-gal staining

10 µm cryosections were fixed for 20 min with 1% glutaraldehyde in Phosphate Buffered Saline Solution (1x PBS) (137 mM NaCl, 2.7 mM KCl, 8.1 mM Na2HPO4, 1.5 mM KH2PO4, pH 7.4). Sections were washed twice for 3–5 min with 1x PBS before being reacted with the X-gal solution (0.2% X-gal, 3.1 mM K3Fe(CN)6, 3.1 mM K4Fe(CN)6, 150 mM NaCl, 1 mM MgCl2, 10 mM Na2HPO4/NaH2PO4) at room temperature. All sections (for a given age and gender) were stained simultaneously to allow comparison. The staining reaction was monitored under a dissecting microscope and stopped for all sections before saturation is reached in any section. Reactions were terminated with 3 3–5 min washes with 1x PBS. The sections were then dehydrated and mounted in 70% glycerol in PBS. Images were obtained on a Zeiss Axioplan II imaging microscope with a Leica DC500 high-resolution camera and the OpenLab imaging software (Improvision, Lexington, MA). All images were captured with the same exposure time and light intensity.

Results

The prevailing view in yeast cells is that the inhibition of GAL4 by GAL80 is mediated by a mechanism by which GAL80 dimers assembles into 2:2 complex with GAL4 dimers bound to UAS sequences (Egriboz et al., 2013). To regulate GAL4 with GAL80 in flies, it is required to express at least as many molecules of GAL80 as GAL4 (Fig. 1A). In order to be applicable to any driver GAL80 would also need to be highly expressed in all tissues. Therefore, tTA is placed under the control of a strong and ubiquitous promoter. Although few ubiquitous promoters have been reported in Drosophila, their expression pattern during aging has not been examined (Ackermann & Brack, 1996; Bond & Davidson, 1986; O’Donnell, Chen & Wensink, 1994). Therefore two versions, pDJ146 and pDJ147, of the Tet-off GAL80 transgenes were constructed (Data S1). Both versions consist of a single P-element containing two promoter-cDNA constructs (promoter-tTA-SV40 transcriptional terminator and TetO-GAL80- β-globin transcriptional terminator). The tTA cDNA is under the control of the tubulin 1α promoter and the actin-5c promoter in the pDJ146 and pDJ147 versions respectively (Fig. 1B). Both of these promoters are expected to drive constitutive ubiquitous expression (Bond & Davidson, 1986; O’Donnell, Chen & Wensink, 1994; Stebbins & Yin, 2001) of the tTA protein and consequently of the TetO-linked GAL80 cDNA. 11 and 5 transgenic strains were obtained with the pDJ146 and the pDJ147 respectively.

It was chosen to test the Tet-Off GAL80 transgenes with a high expressing GAL4 driver since it would be more likely that the findings would also apply to drivers with lower levels of expression. Among the 180 GAL4 and 6 Geneswitch drivers that have been characterized in aging flies, DJ694 and MHC have the highest levels of expression (Poirier et al., 2008; Seroude et al., 2002). Since these two drivers are expressed in muscle tissues the expression of the 24B GAL4 muscle driver was measured in aging flies and found to be highly expressed as well (Fig. S1, Data S1). The pattern of expression of these three drivers was determined in the pre-adult and adult stages with a UAS-GFP reporter (Fig. 2). 24B and MHC are both expressed in the muscle tissues at all stages. Whereas DJ694 is expressed in the adult muscle tissues, in contrast, it is not expressed in the larval and pupal muscles but instead in the oenocytes and salivary glands. The goal of restricting expression solely in the adult muscles would be greatly facilitated by using the DJ694 driver because of the absence of expression in muscles before the adult stage. In the adult, MHC and DJ694 are expressed in the flight, leg and labial muscles, but DJ694 has additional expression in the abdominal muscles (Poirier et al., 2008; Seroude et al., 2002). While 24B is also mainly expressed in muscles, the distribution is less ubiquitous as the animal ages and become mainly restricted to a subset of the indirect wing muscles (Fig. S1). Therefore, DJ694 was selected as the driverused to test the Tet-off GAL80 transgenes since it has the dual advantage of undetectable expression in muscles during the pre-adult stages and high, ubiquitous expression in the adult muscles.

Figure 2 Expression pattern of the DJ694, 24B, and MHC drivers visualized with a UAS-GFP reporter.

GAL80 repression of pre-adult UAS expression

UAS-grim lethality test

In order to assess the ability of the Tet-off GAL80 transgenes to repress the expression of a UAS transgene, lethality tests were performed in presence of a UAS-grim reporter. The Grim reporter gene encodes a strong pro-apoptotic factor, thus the lethality across development (embryonic, larval, pupal) can be scored to assess the repression ability of GAL80. The UAS-grim lethality test facilitates the determination of the developmental period during which GAL4 activity is not inhibited. Among several hundred GAL4 drivers covering a wide range of expression levels and localizations, 95% displayed partial or complete lethality with this test (Seroude, 2002; Seroude et al., 2002).

When expression of UAS-Grim is driven by DJ694 50–60% of the eggs hatch and no adults are ever recovered. Genotypes with 1 copy of the DJ146 transgene added displayed ∼60–78% hatching, ∼25–60% pupation and less than 2% of uneclosed or partially eclosed adults were obtained but none were viable (Fig. S13). Comparison between all DJ146 insertions revealed no significant difference in survival at any stage with the exception of the DJ1081 insertion (Table S2). DJ1081 showed a significantly higher rate of eclosion that did not decrease in presence of tetracycline (Fig. S14) and was excluded from subsequent experiments. It was therefore possible to group genotypes (Fig. 3, Table S3). The addition of one DJ146 transgene showed pupation rates that are significantly higher than the negative control but remained significantly lower than the positive control. Genotypes with 1 copy of the DJ147 transgene resulted in ∼51–75% hatching, ∼13–23% pupation and less than 2% of adults that were not viable (Fig. S15). Like DJ146 the comparison between all DJ147 insertions showed no significant difference in survival at any stage (Table S2). The addition of a DJ147 transgene revealed a significant increase in the number of pupae relative to the negative control, however the improvement was still significantly lower than the positive control. These findings indicate that neither one copy of DJ147 nor DJ146 is sufficient to recover from grim-induced death to a level comparable to animals that do not express grim.

Figure 3 Developmental repression of DJ694 GAL4 transcriptional activity by Tet-off GAL80 transgenes in the absence of inducer, as measured by UAS-grim induced lethality.

The y-axis shows the percentage of individuals alive relative to the previous stage. The x-axis shows the type of Tet-off GAL80 transgene and copy number. %L1 (white): percentage of eggs that hatched. %P (light grey): percentage of L1 larvae that pupated. %A (dark grey): percentage of pupae that yielded adults. Error bars represent ±2SD. P values are provided in Tables S2 and S3.

Genotypes with two copies of DJ146 yielded ∼53–97% hatching, ∼44–78% pupation and ∼0–43% eclosion (Fig. S17). The pupation rates are significantly higher than the negative control for all insertion combinations as well as most hatching rates. Only two genotypes produced significantly fewer larvae than the positive control (2 copies DJ1077, DJ1077 + DJ1080) and all genotypes produced significantly less pupae. 8 out of 10 genotypes showed a significant improvement of the number of adults but it remained lower than the positive control. Genotypes with two copies of DJ147 showed ∼55–92% hatching, ∼19–85% pupation and ∼0–41% eclosion (Fig. S19). The pupation rates are significantly higher than the negative control for all insertion combinations as well as most hatching rates. Hatching (3.3 + 4) and pupation (two copies 2.1) rates were comparable to the positive control for only one genotype. Five out of seven genotypes showed a significant increase in eclosion but it remained lower than the positive control. The addition of a second copy of either transgene further reduced lethality but still failed to reach wild-type levels. The addition of three or four copies of DJ146 showed a significant improvement in pupation and eclosion in all genotypes relative to the negative control (Figs. S23 and S27). It is evident that the number of copies of a GAL80 insertions is correlated with a decrease in grim-induced lethality but it is also dependent on the developmental stage and the kind of GAL80 transgene, and the addition of a fourth copy does not lead to further improvement. This observation suggested that in neither version of the transgene is GAL80 expressed ubiquitously and/or uniformly at all stages. This possibility was examined by testing if genotypes with combinations of DJ146 and DJ147 have a cooperative effect (Fig. S21). Indeed, the combination of one copy of DJ146 and one copy of DJ147 showed a significant improvement in eclosion compared to two copies of either transgene (Table S3). This synergistic effect was confirmed with three and four copies combinations that always showed a significant improvement in comparison with three or four DJ146 copies (Figs. S25 and S29).

Overall it appears that increasing GAL80 copy number does improve survival during all developmental stages but the most complete inhibition of lethality required the presence of the two kind of transgenes.

UAS-lacZ expression

The recovery of viable animals with the UAS-grim test does not prove that the expression of the UAS promoter has been completely abolished and does not exclude the possibility that grim is expressed in cells refractory to apoptosis. Therefore, the expression of a UAS-lacZ reporter was examined during development (Fig. 4, Tables S4–S8).

Figure 4 Developmental repression of DJ694 GAL4 transcriptional activity by Tet-off GAL80 transgenes in the absence of inducer, as measured by UAS-lacZ.

The y-axis shows the lacZ-encoded ß-galactosidase activity (ΔmOD562nm/min/µg protein) normalized to the level seen with the positive control (no GAL80 transgene). The x-axis shows the type of Tet-off GAL80 transgene and copy number. L1 (white): first instar larvae; L3 (light grey): third instar larvae; EP (dark grey): early pupae; LP (black): late pupae. Error bars represent the range of experimental values. P values are provided in Table S8.

In agreement with the UAS-grim assay, the presence of GAL80 significantly repressed the expression of the reporter during at least two of the four developmental stages tested (Tables S4–S7), but the repressive ability was dependent on the copy number and the GAL80 variant (Table S8). A single copy of DJ146 resulted in expression levels significantly lower than a single copy of DJ147 in L3 and early pupae extracts. The addition of a second DJ146 copy did not significantly decrease expression at any stage while the addition of a second DJ147 copy did in L1 and early pupae. The addition of a third DJ146 copy did significantly reduce expression across all stages whereas a third DJ147 copy only had a significant effect in the late pupae extract.

The results provided additional evidence that the two kind of GAL80 transgenes have synergistic effects. The presence of one copy DJ146 and one copy DJ147 significantly decreased expression compared to genotypes with two copies of the same kind of transgene. Similarly three copies combinations (2x DJ146 + DJ147; DJ146 + 2x DJ147) showed significant differences with three copies of the same transgene. Additionally four copies combinations were significantly different from four DJ146 copies.

The UAS-lacZ reporter revealed low expression levels with two copies of each kind of GAL80 transgenes present. The localization of the expression was therefore investigated in wandering L3 larvae and late pupae (Fig. 5). With one copy of either transgene, salivary glands and oenocytes are stained confirming that GAL80 does not completely inhibit GAL4 activity in those tissues. With two copies of each transgene, staining is no longer observed except in a single group of oenocytes at the posterior tip of the pupal abdomen.

Figure 5 Distribution of DJ694-mediated UAS-lacZ expression in the absence of inducer visualized by X-gal staining of L3 larvae (A, C, E, G, I, K) and late pupae (B, D, F, H, J, L).

(A, B) No GAL80: positive control carrying the DJ694 driver and the UAS-lacZ reporter. (C, D) 1 X DJ146: (DJ694/UAS-lacZ; DJ1077/+); (E, F) 1 X DJ147 (DJ694/UAS-lacZ; 3.3/+); (G, H) 2 X DJ146 (DJ694/UAS-lacZ; DJ1077/DJ1079); (I, J) 1 X DJ146 + 1 X DJ147 (DJ694/UAS-lacZ; 3.3, DJ1077/+); (K, L) 2 X DJ146 + 2 X DJ147 (DJ694/UAS-lacZ; 3.3, DJ1077). Arrows: salivary glands. Arrow heads: oenocytes.

Close examination of the controls provides several additional evidence that both GAL80 transgenes are not equivalent despite both expressing tTA under the control of a ubiquitous promoter. Whereas both larval salivary glands are stained similarly with either GAL80 transgene, most oenocytes are positive with DJ146 while only a subset is with DJ147. While most pupal oenocytes are positive with both transgenes, the expression in the salivary glands is barely detectable or undetectable with DJ147. Finally, the addition of a second DJ146 transgene decreased larval expression in most oenocytes but not in the salivary glands. In contrast adding a copy of DJ147 to DJ146 resulted in a decrease in both locations. In pupae having two identical or a combination of transgenes reduced the expression in the salivary glands but only the combination decreased the staining of the oenocytes.

tetO-lacZ expression

To investigate the differences observed between the two kind of GAL80 transgenes, the expression of tTA was examined directly with a tetO-lacZ reporter. The ubiquitous promoters controlling tTA indeed yielded high and widely distributed signal that required short staining times to reveal qualitative differences. Both DJ146 and DJ147 transgenes revealed lacZ activity in the muscles, trachea, oenocytes and Malpighian tubules of larvae (Fig. 6). However, obvious differences can be seen in the larval digestive and nervous systems. Whereas DJ146 expresses tTA ubiquitously in the digestive tract, brain lobes and ventral chord, DJ147 does not stain areas of the gut, and the central nervous system. While the expression in the digestive system is similar for both transgenes in the pupal and adult stages, expression in the central nervous remains undetectable with DJ147.

Figure 6 Distribution of tTA-mediated TetO-lacZ expression with the DJ146 (A–D) and DJ147 (E–H) transgenes visualized by X-gal staining of dissected L3 larvae (A, B, C, E, F, G) and cryosectioned late pupae (D, H).

(A, E) musculature, trachea and oenocytes (after removal of digestive and nervous systems). (B, F) digestive system before removal. (C, G) central nervous system. DJ146: tetO-lacZ/+; DJ1077/+; DJ147: tetO-lacZ/+; 2.1/+. b, brain; vc, ventral cord.

Adult expression

Once an almost complete inhibition of GAL4 activity can be achieved during development, the repression and induction properties of the Tet-off GAL80 transgenes was examined in adult flies carrying two copies of DJ146, two copies of DJ147, the DJ694 driver and the UAS-lacZ reporter.

Throughout-life induction

Individuals carrying the UAS-lacZ reporter alone or in combination with the DJ694 driver served as negative and positive controls, respectively. The level of lacZ expression in the experimental and control populations was measured with a CPRG assay at different ages spanning ∼70% of the life span. At each age, four concentrations of inducer (0, 1, 10 and 100 µg/ml tetracycline) were tested simultaneously (Fig. 7). Comparisons between experimental and control animals were performed to determine the repressive ability of GAL80 (Table S9). In both sexes the presence of the GAL80 transgenes significantly reduced lacZ expression but never dropped it to the level of the negative control. The repression of GAL4 activity is impaired in older animals. In males, up to 20 days of age, expression levels are less than 5% of the positive control. At later ages, the levels increased to reach 40% of the positive control at 40 days of age. Females displayed less repression than males. The expression level is only reduced to 10% of the positive control at two days and to 20–25% up to 20 days. At later ages, the levels also increased to more than 50% by 40 days. The lack of repression at advanced ages could be the result of a decreased expression of the ubiquitous promoters controlling tTA or a decreased ability of tTA to activate the tetO promoter leading to reduced GAL80 expression. Alternatively, the ability of GAL80 to bind and inhibit GAL4 may decline. To distinguish between these hypotheses, the level of tTA expression mediated by DJ146, DJ147 or both transgenes was measured across the life span with the tetO-lacZ reporter gene (Fig. 8). In both males and females, the expression either increased or remained stable across life for the DJ146 and DJ147 GAL80 transgenes respectively. As expected, the expression level is higher with both transgenes but the profile remains similar. These results indicate that a decline of tTA expression or activity cannot account for the increased GAL4 activity at advanced ages in absence of inducer and support the hypothesis that the repressive activity of GAL80 is declining.

Figure 7 Adult induction of DJ694 GAL4 transcriptional activity with different inducer concentrations as measured by UAS-lacZ in adult males (A) and females (B) with two copies of each kind of Tet-off GAL80 transgene (DJ694/UAS-lacZ; 3.3, DJ1077).

The y-axis shows lacZ-encoded ß-galactosidase activity (ΔmOD562nm/min/fly) normalized to the level seen with the positive control (no GAL80 transgene). The x-axis shows age in days. Error bars represent ±2SD (n = 5 per tetracycline treatment per age).

Figure 8 DJ146 and DJ147 tTA transcriptional activity throughout adulthood (A: males, B: females) measured using a tetO-lacZ reporter.

The y-axis shows lacZ-encoded ß-galactosidase activity (ΔmOD562nm/min/fly). The x-axis shows age in days. Error bars represent ±2SD (n = 5 per genotype per age).

Next the inducibility was examined with the same methodology used for the Geneswitch system (Poirier et al., 2008). The different inducer concentrations were compared to each other to determine the number of inducing concentrations and how many different levels of expression were elicited over the various ages tested (Table S10). Two days of treatmentwith any concentration was insufficient to induce lacZ expression in either sex. From five to 20 days, all concentrations induced in both sexes but resulted in two distinct levels of expression in males instead of a single in females. At 30 days, the number of inducing concentrations drops and a single level of expression is seen. These observations raised the concern that the inducibility changes with age and led to investigate if the age at which the tetracycline treatment is started influences the ability to induce GAL4 activity (Data S1).

Throughout-life induction: tissue localization

To characterize where GAL4 activity is induced, flies treated with 0 or 100 µg/ml of tetracycline were sectioned and stained with X-Gal at 10 days and 30 days of age (Figs. 9 and 10). In the absence of GAL4, no expression is detected in any tissue of either sex. Weak signal may be observed in some areas of the male digestive system but is not consistent between individuals within the same genotype and treatment. In 10 days old animals with four copies of GAL80, lacZ expression is strongly reduced compared to the positive control animals (DJ694). However, in agreement with the CPRG assay, some expression remains in the thoracic flight muscles and the head, and it is higher in the females than the males. Animals fed tetracycline show induction of lacZ expression in the thoracic, abdominal, labial and leg muscles, mirroring the expression pattern seen in the positive control. In 30 days old animals, higher levels of uninduced expression is observed in the thoracic flight muscles in both sexes. At this age, uninduced expression is additionally seen in the male abdominal muscles. Animals fed tetracycline still show induction in the abdominal, labial and leg muscles but none (females) or almost none (males) is visible in the thoracic muscles.

Figure 9 Distribution of DJ694 GAL4-mediated UAS-lacZ expression visualized by X-gal staining of cryosections of 10 (A–F) and 30 (G–L) days old adult males in presence (100 µg/ml tetracycline) (B, D, F, H, J, L) or absence of the inducer (A, C, E, G, I, K).

(A, B, G, H) w 1118: negative control (UAS-lacZ/+; 3.3, DJ1077/+). (C, D, I, J) DJ694: positive control (DJ694/UAS-lacZ). (E, F, K, L) GAL80: experimental animals (DJ694/Bg2; 3.3, DJ1077). Several pictures of the same section were stitched together to generate whole-fly views. am, abdominal wall muscle; fm, flight muscle; lgm, leg muscle; lm, labial muscle.

Figure 10 Distribution of DJ694 GAL4-mediated UAS-lacZ expression visualized by X-gal staining of cryosections of 10 (A–F) and 30 (G–L) days old adult females in presence (100 µg/ml tetracycline) (B, D, F, H, J, L) or absence of the inducer (A, C, E, G, I, K).

(A, B, G, H): w 1118: negative control (UAS-lacZ/ +; 3.3, DJ1077/ +). (C, D, I, J) DJ694: positive control (DJ694/UAS-lacZ). (E, F, K, L) GAL80: experimental animals (DJ694/Bg2; 3.3, DJ1077). Several pictures of the same section were stitched together to generate whole-fly views. am, abdominal wall muscle; fm, flight muscle; lgm, leg muscle; lm, labial muscle.

GAL4 driver survey

To test the effect of Tet-off GAL80 transgenes on GAL4 drivers other than DJ694, a panel of 26 drivers was tested with the UAS-grim lethality assay in presence of one copy of DJ146, one copy of DJ147, or one copy of both DJ146 and DJ147. The parental genotypes, number of experiments performed, and scores from each plate is provided in Table S13. Repression of individuals carrying a copy of each DJ146 and DJ147 were compared to individuals that did not carry any GAL80 transgenes (Table 1). Of the eleven drivers (42.3%) that show massive embryonic lethality (less than 5% hatching), nine (81.8%) display significant improvement in hatching with the GAL80 transgenes. Of the 15 drivers that cause 50% or less lethality, one displayed improved hatching. Of the 25 drivers for which L1 larvae are obtained in presence of GAL80, 18 (72%) showed higher pupation rate. Of the 18 drivers for which pupae are obtained, 6 (33.3%) had higher eclosion rate.

In agreement with the UAS-grim assay the addition of DJ146 and DJ147 GAL80 transgenes significantly repressed the reporter expression in the UAS-lacZ assay (Table 2). GAL80 decreased lacZ expression in the L1 larvae for all the drivers with the exception of DJ710 (96%). However this driver did not reduce the hatching rate in the UAS-grim assay, which indicates that the driver has little or no expression in the embryo. In the L3 larvae, GAL80 decreased expression for all of the drivers (100%). In the early pupae, lacZ expression was decreased in all drivers except for ELAV (96%). In the late pupae, the addition of GAL80 decreased lacZ expression in all drivers except ELAV and 24B (93%). As one would expect, a much higher proportion of drivers are suppressed with the CPRG assay.

Discussion

In order to control the expression of a gene of interest during the adult portion of the fly life cycle, an inducible gene expression system must be able to completely repress transgene expression during development. Two UAS reporters were used to assess the repression ability of the Tet-off GAL80 transgenes. The UAS-grim reporter assesses expression among cells expressing GAL4 that are required for organism survival. The UAS-lacZ reporter examines expression in all cells where GAL4 is expressed. The UAS-grim reporter facilitated the identification of insufficiently repressed developmental stages and, in combination with the UAS-lacZ reporter, extended our knowledge of the relationship between the level and localization of grim expression, and its biological effect. The addition of a second copy of the DJ147 transgene resulted in significant higher recovery of adults but only significantly reduced lacZ expression during the early stages of pupation, indicating that GAL4 must be expressed in early pupal cells that are essential to obtain adults and a single copy of DJ147 is unable to produce enough GAL80 in those cells. The addition of a second copy of the DJ146 transgene did not reduce lacZ expression but did improve the recovery of pupae. The localization of lacZ expression in 2x DJ146 L3 larvae showed reduced staining in the oenocytes that could be detected with the grim lethality test but not with the CPRG assay.

The repression of GAL4 activity during development is dependent on the number of Tet-off GAL80 transgenes present. Although one copy of either GAL80 transgene significantly improved the recovery of pupae, neither completely abolished GAL4 activity. The level of GAL4 expression measured in L3 larvae ranged between 1.1 and 6.5 ΔmOD/min/µg (average: 3.2 ± 1.5, n = 38). The levels of expression of GAL80 mediated by the Tet-off GAL80 transgenes were more than 15 (DJ147) and 40 (DJ146) times higher than GAL4 (Fig. S32). Since tTA is widely expressed in L3 but not uniformly distributed, and the recovery of pupae increased with a second copy of either GAL80 transgene, the lack of repression is most likely due to insufficient GAL80 expression in some cells expressing GAL4 rather than a cell-specific inability to produce active GAL80 molecules. Three or four copies did not further increase the pupation rate despite significantly reducing lacZ activity suggesting that GAL80 expression remains insufficient in approximatively one third of the animals or that neither GAL80 transgene is expressed in some cells that are not strictly required for the survival of every single individual or that have a variable grim susceptibility between individuals.

Table 1 Survey of GAL4 drivers with the UAS-grim lethality test.

GAL4 driver	Hatching	Pupation	Eclosion	p-value	
	−GAL80	+GAL80	−GAL80	+GAL80	−GAL80	+GAL80				
	AVE.	SD	AVE.	SD	AVE.	SD	AVE.	SD	AVE.	SD	AVE.	SD	Hatching	Pupation	Eclosion	
da-GAL4	0.0	0.0	0.0	0.0	0.0	0.0	0.0	0.0	0.0	0.0	0.0	0.0	∼1	UD	UD	
tubGAL4	0.0	0.0	7.0	6.7	0.0	0.0	0.0	0.0	0.0	0.0	0.0	0.0	6.8E−02	∼1	UD	
24B	0.0	0.0	31.0	10.0	0.0	0.0	0.0	0.0	0.0	0.0	0.0	0.0	4.5E−07	∼1	UD	
ELAV	0.0	0.0	52.0	22.6	0.0	0.0	0.0	0.0	0.0	0.0	0.0	0.0	1.2E−03	∼1	UD	
DJ752	0.0	0.0	62.5	21.2	0.0	0.0	40.1	18.6	0.0	0.0	0.0	0.0	1.8E−04	1.8E−03	∼1	
DJ761	0.0	0.0	68.0	10.5	0.0	0.0	0.0	0.0	0.0	0.0	0.0	0.0	1.7E−07	∼1	UD	
DJ755	0.0	0.0	70.5	16.7	0.0	0.0	48.8	7.9	0.0	0.0	0.0	0.0	9.0E−06	2.7E−07	∼1	
DJ756	0.0	0.0	61.0	14.8	0.0	0.0	46.7	8.8	0.0	0.0	0.0	0.0	1.1E−05	1.2E−06	∼1	
DJ1007	0.0	0.0	59.5	10.4	0.0	0.0	29.5	8.7	0.0	0.0	0.0	0.0	5.5E−07	6.0E−05	∼1	
DJ1040	0.0	0.0	71.0	9.7	0.0	0.0	55.0	17.7	0.0	0.0	0.0	0.0	5.8E−08	1.2E−04	∼1	
D42	4.5	5.0	45.0	7.6	0.0	0.0	23.6	14.6	0.0	0.0	0.0	0.0	5.2E−09	4.4E−04	∼1	
DJ817	51.0	10.0	70.0	13.2	0.0	0.0	30.0	7.7	0.0	0.0	0.0	0.0	3.0E−02	1.9E−05	∼1	
DJ634	56.0	11.8	55.5	11.8	0.0	0.0	7.2	8.8	0.0	0.0	0.0	0.0	9.3E−01	3.6E−02	∼1	
DJ715	68.5	12.9	31.0	7.0	0.0	0.0	0.0	0.0	0.0	0.0	0.0	0.0	4.4E−06	∼1	UD	
DJ695	78.0	7.7	65.0	26.2	0.0	0.0	0.0	0.0	0.0	0.0	0.0	0.0	3.6E−01	∼1	UD	
DJ946	78.0	13.7	65.0	11.7	0.0	0.0	49.5	8.9	0.0	0.0	34.8	38.3	6.0E−02	2.5E−10	2.2E−02	
DJ785	79.0	5.0	79.5	12.9	0.0	0.0	55.6	12.5	0.0	0.0	96.1	4.5	9.4E−01	5.6E−06	1.4E−12	
DJ849	79.5	13.1	65.0	7.3	56.3	6.7	63.5	20.9	86.5	15.1	98.6	3.9	1.6E−02	3.7E−01	4.5E−02	
ddc-GAL4	82.0	10.0	79.5	9.9	0.0	0.0	0.0	0.0	0.0	0.0	0.0	0.0	6.2E−01	∼1	UD	
DJ1027	84.0	7.3	78.0	16.0	41.8	3.8	58.1	15.1	100	0.0	50.2	24.3	5.0E−01	6.5E−02	2.5E−03	
DJ628	84.0	15.1	59.0	22.0	0.0	0.0	34.3	26.6	0.0	0.0	0.0	0.0	1.9E−02	2.7E−03	∼1	
DJ646	86.5	14.6	82.0	9.8	20.8	13.0	66.7	13.8	24.2	32.3	97.0	4.2	4.8E−01	7.7E−06	1.9E−05	
GAL4.109	88.0	3.3	68.5	11.6	0.0	0.0	6.3	7.9	0.0	0.0	0.0	0.0	9.1E−03	1.5E−01	∼1	
Nrv2-GAL4	92.5	6.9	79.5	7.5	0.0	0.0	49.2	13.7	0.0	0.0	63.3	13.9	2.9E−03	7.4E−08	3.9E−09	
DJ710	97.0	3.8	70.5	21.5	0.0	0.0	61.1	26.0	0.0	0.0	27.7	14.4	3.8E−02	1.0E−03	3.8E−03	
cha-GAL4	98.0	2.1	86.0	8.6	0.0	0.0	1.7	3.5	0.0	0.0	0.0	0.0	1.8E−03	1.9E−01	∼1	
Notes.

AVE average percent of previous stage

SD standard deviation

−GAL80 negative control genotypes (driver + UAS-grim)

+GAL80 experimental genotypes (driver + UAS-grim + 146 and 147 GAL80 transgenes)

p-values represent comparisons between animals with or without GAL80, significant values (p < 0.05) are highlighted in grey. UD, unable to determine (no data).

Table 2 Survey of GAL4 drivers with UAS-lacZ.

Data presented are the average specific activity (ΔmOD562 nm/min/µg protein) plus or minus the standard deviation (n = 5).

GAL4 driver	−GAL80	+GAL80	p-value	
	L1	L3	EP	LP	L1	L3	EP	LP	L1	L3	EP	LP	
da-GAL4	9.5 ± 1.0	84.6 ± 18.1	79.1 ± 3.7	134.3 ± 17.4	1.3 ± 0.2	23.7 ± 8.1	24.7 ± 8.5	21.9 ± 7.6	6.9E−08	2.4E−03	1.1E−04	4.1E−05	
tub-GAL4	12.1 ± 3.6	136.3 ± 30.4	113.5 ± 14.5	165.3 ± 15.9	1.4 ± 0.3	30.1 ± 3.6	35.2 ± 15.8	26.6 ± 10.7	7.7E−05	2.7E−05	1.9E−05	1.1E−07	
24B	3.1 ± 0.6	22.2 ± 2.9	29.3 ± 5.6	34.1 ± 6.2	0.3 ± 0.2	13.1 ± 5.3	11.8 ± 4.7	36.8 ± 5.7	2.8E−06	5.0E−03	3.5E−04	2.4E−01	
ELAV	1.1 ± 0.1	0.5 ± 0.1	0.4 ± 0.1	1.2 ± 0.3	0.3 ± 0.1	0.4 ± 0.1	0.4 ± 0.1	1.1 ± 0.2	4.6E−08	2.0E−03	1.1E−01	2.2E−01	
DJ752	2.2 ± 0.7	15.6 ± 5.7	20.8 ± 4.3	9.1 ± 1.5	0.2 ± 0.1	2.4 ± 1.2	1.3 ± 0.7	1.7 ± 0.3	7.3E−05	5.2E−04	4.0E−06	0.0E+00	
DJ761	2.5 ± 0.6	83.8 ± 11.1	47.1 ± 8.6	33.8 ± 4.1	1.4 ± 1.1	8.3 ± 2.6	2.7 ± 0.5	5.4 ± 1.0	3.4E−02	2.2E−07	1.5E−06	1.7E−07	
DJ755	4.3 ± 1.4	12.9 ± 3.3	16.6 ± 6.1	12.0 ± 4.1	0.6 ± 0.2	2.0 ± 0.8	2.3 ± 0.2	1.7 ± 0.3	2.4E−04	4.5E−05	1.6E−04	8.5E−05	
DJ756	5.7 ± 1.8	18.1 ± 4.8	19.1 ± 6.4	8.7 ± 1.9	0.3 ± 0.1	1.9 ± 1.0	1.7 ± 1.1	2.3 ± 0.9	3.2E−04	3.8E−05	1.7E−04	5.9E−05	
DJ1007	5.6 ± 2.1	7.8 ± 4.1	5.5 ± 1.3	5.5 ± 1.8	1.6 ± 0.9	0.8 ± 0.4	1.6 ± 0.6	1.6 ± 0.4	2.0E−03	2.5E−03	1.3E−04	9.2E−04	
DJ1040	5.9 ± 1.1	4.8 ± 2.6	8.0 ± 2.0	5.1 ± 0.8	0.8 ± 0.4	0.9 ± 0.4	1.9 ± 0.2	2.6 ± 1.3	3.1E−06	5.7E−03	6.8E−05	2.6E−03	
D42	2.0 ± 0.2	11.9 ± 2.8	6.4 ± 3.4	2.4 ± 0.4	0.3 ± 0.1	1.1 ± 0.7	0.8 ± 0.4	0.6 ± 0.1	4.7E−08	1.6E−05	4.1E−03	8.6E−06	
DJ817	3.7 ± 0.8	9.2 ± 4.8	10.5 ± 1.2	4.1 ± 1.8	0.2 ± 0.1	0.9 ± 0.5	0.6 ± 0.5	0.7 ± 0.7	5.7E−06	2.6E−03	7.7E−08	1.3E−03	
DJ634	0.3 ± 0.1	20.4 ± 4.1	23.5 ± 5.5	48.3 ± 10.1	0.1 ± 0.1	2.7 ± 0.7	1.1 ± 0.8	0.6 ± 0.2	2.1E−04	5.6E−06	8.7E−06	2.7E−06	
DJ715	2.7 ± 0.4	23.8 ± 0.9	17.7 ± 5.4	15.2 ± 0.1	1.3 ± 2.1	1.9 ± 0.1	2.1 ± 0.1	0.4 ± 0.1	8.7E−02	9.7E−06	1.8E−04	5.0E−05	
DJ695	0.2 ± 0.1	7.1 ± 2.1	9.7 ± 5.2	2.8 ± 1.1	0 ± 0	1.1 ± 0.3	0.5 ± 0.3	0.3 ± 0.1	3.0E−05	9.1E−05	2.3E−03	3.9E−04	
DJ946	3.3 ± 1.0	24.8 ± 13.2	7.7 ± 2.6	7.9 ± 1.4	0.1 ± 0.1	0.6 ± 0.4	0.3 ± 0.1	0.2 ± 0.1	4.1E−05	1.7E−03	1.2E−04	8.8E−07	
DJ785	1.2 ± 0.6	17.3 ± 2.2	17.1 ± 7.1	2.6 ± 1.6	0.4 ± 0.4	2.4 ± 1.7	0.4 ± 0.4	0.2 ± 0.1	1.0E−02	9.4E−07	4.0E−04	4.3E−03	
DJ849	0.4 ± 0.1	19.3 ± 5.2	12.1 ± 1.8	1.2 ± 0.4	0.1 ± 0.1	1.5 ± 0.5	0.6 ± 0.3	0.2 ± 0.1	8.7E−06	3.2E−05	3.2E−07	3.0E−04	
ddc-GAL4	1.2 ± 0.4	2.2 ± 1.1	5.9 ± 3.3	9.2 ± 4.4	0.2 ± 0.1	0.1 ± 0.1	0.2 ± 0.1	2.7 ± 1.6	1.6E−04	1.4E−03	3.1E−03	7.3E−03	
DJ1027	2.9 ± 2.1	8.6 ± 8.0	8.6 ± 5	5.1 ± 4.8	0.7 ± 0.3	0.3 ± 0.4	2.3 ± 0.4	1.6 ± 1.5	6.7E−02	2.5E−02	1.1E−02	8.3E−02	
DJ628	0.9 ± 0.6	27.7 ± 6.5	11.9 ± 5.9	36.5 ± 22.1	0 ± 0	5.2 ± 1.5	3.4 ± 2.1	2.6 ± 1.5	3.9E−03	3.4E−05	8.2E−03	4.4E−03	
DJ646	0.4 ± 0.1	12.4 ± 3.2	7.9 ± 3.3	0.7 ± 0.3	0.1 ± 0.1	0.5 ± 0.3	0.2 ± 0.1	0.1 ± 0.1	1.4E−04	1.8E−05	3.7E−04	1.4E−03	
GAL4.109	0.5 ± 0.1	12.5 ± 2.2	8.3 ± 2.6	11.9 ± 2.1	0.1 ± 0.1	0.2 ± 0.1	0.5 ± 0.2	0.8 ± 0.2	2.2E−06	7.3E−07	9.0E−05	1.0E−06	
Nrv2-GAL4	0.3 ± 0.1	1.0 ± 0.2	1.5 ± 0.3	6.0 ± 1.3	0.1 ± 0.1	0.2 ± 0.1	0.2 ± 0.1	1.6 ± 0.7	2.3E−03	4.7E−05	5.9E−06	9.0E−05	
DJ710	0.5 ± 0.1	11.2 ± 5.4	6.8 ± 5.8	0.5 ± 4.7	0.4 ± 0.2	0.1 ± 0.6	0.1 ± 1.0	0.1 ± 0.1	1.0E−01	2.1E−09	1.2E−02	1.2E−05	
cha-GAL4	0.5 ± 0.1	0.7 ± 0.2	0.3 ± 0.1	1.4 ± 0.4	0.2 ± 0.1	0.2 ± 0.1	0.2 ± 0.1	0.8 ± 0.2	1.1E−03	4.9E−05	1.8E−02	7.7E−03	
Notes.

−GAL80 positive control genotypes (driver + UAS-lacZ); +GAL80, experimental genotypes (driver + UAS-grim + 3.3,DJ1077)

L1 first instar larvae

L3 third instar larvae

EP early pupae

LP late pupae

p-values represent comparisons between animals with or without GAL80, significant values (p < 0.05) are highlighted in grey.

The repression of GAL4 activity during development is also dependent on the type of Tet-off GAL80 transgenes present. One copy of either GAL80 transgene failed to enable the recovery of viable adults. The GAL4 expression pattern extends during the pupal stages and the level ranged between 6 and 49 ΔmOD/min/µg (average: 19.1 ± 11.5, n = 37) in late pupae. Meanwhile GAL80 expression did not increase as much as GAL4 reducing the difference to 5- to 8-fold for DJ147 and DJ146 respectively (Fig. S32). It is therefore not surprising that three copies of DJ146 are required to obtain similar recovery of adults and pupae. However, the recovery of adults was significantly better with a copy of each type of GAL80 transgene than with two copies of the same type. Since only the heterologous combination of transgenes affected the expression of the UAS-lacZ reporter in the pupal oenocytes, the synergistic effect of the combination suggests that DJ147 is expressed higher than DJ146 in the pupal oenocytes despite a ∼30% lower global expression (Fig. S32). Additionally, it is also possible that the synergistic effect results from some non-overlapping of the GAL80 expression pattern between DJ146 and DJ147, that once combined match the GAL4 expression pattern. Although two promoters believed to be ubiquitously and constitutively expressed (Drosophila tubulin 1α promoter (DJ146) and the Drosophila actin-5c promoter (DJ147)) were used to drive tTA expression, neither of the GAL80 transgenes showed homogenous TetO-LacZ expression and DJ147 was not expressed in the central nervous system.

Several issues complicate the interpretation of the results. Which cells, tissue or organs whose dysfunction or ablation would cause the death of the organism at any stage of the life cycle have not been extensively investigated. Some tissues are indeed expected to be more detrimental than others and to be more or less sensitive to grim-induced cell death. It is also unknown for most tissue what proportion needs to be preserved for organism survival. The necessity for a particular tissue to be present and the amount of functionality required may change depending on the developmental stage or the age of the animal. Fortunately, the effect on pupation and eclosion rate of the ablation or improperly developed oenocytes and salivary glands have been reported. The BthD gene is specifically expressed in the developing embryonic salivary glands (Kwon et al., 2003). The reduction of BthD expression by RNAi severely disturbed the morphogenesis of the gland, yet lethality was observed only after pupation. It is therefore unlikely that the reduced pupation rate in the UAS-grim lethality test results from a disturbance of the salivary glands. Oenocytes are required for larval and pupal development (Makki, Cinnamon & Gould, 2014). The ablation of larval oenocytes lead to polyphasic lethality associated with developmental arrest and changes in lipid metabolism (Gutierrez et al., 2007; Parvy et al., 2012). Remarkably the larval lethality can be avoided and normal pupation rate can be restored if only half of the larval oenocytes are eliminated (Gutierrez et al., 2007). However the resulting pupae are lethal with a high frequency failing to emerge from the pupal case. The eclosion phenotypes is also seen with null alleles of the Cyp4g1 oenocytes-specific gene or with spidey oenocytes-specific knockdown (Chiang et al., 2016; Gutierrez et al., 2007). Among the genotypes that displayed eclosion rate significantly different from the positive control, failing pupae showed a strikingly similar eclosion phenotype: metamorphosis appeared to have completed normally, the pupal case has opened but the pharate died. Since only the combinations of DJ146 and DJ147 clearly reduced the expression of the UAS-lacZ reporter in the pupal oenocytes, it is therefore very likely that the pupal lethality is due to insufficient GAL4 repression in the oenocytes.

The regulation of the UAS promoter by four copies of the Tet-off GAL80 transgenes was examined across adulthood. Measurements across age are extremely important sinceaging often affects the expression profile and sometimes the expression pattern of GAL4 drivers (Seroude et al., 2002). Moreover, feeding decreases with age and could significantly reduce the amount of inducer ingested (Carey et al., 2006). The methodology previously used to test GeneSwitch GAL4 drivers allowed for the characterization of the inducibility of Tet-off GAL80 transgenes, their ability to translate different inducer concentrations into distinct expression levels and their aptitude to maintain induction (Poirier et al., 2008). The inducibility is comparable between males and females. All inducer concentrations in both sexes showed expression levels that are significantly higher than the untreated control after five but not two days of induction. It took 20 days for the expression levels to be undistinguishable from the control lacking the GAL80 transgenes with the exception of the lowest inducer concentration in males. The three inducer concentrations yielded similar expression profiles but generated two significantly distinct levels of expression only in males. The maintenance of the induction is clearly affected by age since the 40 days old males and females do not show significant differences with the untreated controls. The repressive ability of GAL80 is also affected by age but cannot account for the loss of maintenance because at least 50% of the expression is still repressed in the oldest animals. It is more likely that age impairs the inactivation of tTA by the inducer or the turnover of GAL80.

The length of time to induce and the loss of repression with age of the Tet-off GAL80 transgenes remain an issue for studies that take place over a short time or that use old animals. The loss of repression can be addressed by testing different promoters to drive tTA expression and the timing of the induction may be different for a different GAL4 driver. A small survey of a variety of GAL4 drivers revealed that they are indeed differently affected by the GAL80 transgenes (Tables 1 and 2). The GAL80 transgenes will need further optimization to reach the absolute repression that a geneticist requires to reach unambiguous conclusions about the effect of a gene of interest at any time in any anatomical location.

Supplemental Information

Supplemental Information 1 Supplemental data

Supplemental text, figures and tables (datasets and statistics).

Click here for additional data file.

Supplemental Information 2 pDJ146/pDJ147 sequences

pDJ146.nucl: pDJ146 sequence (macVector format)

pDJ146.gb: pDJ146 sequence (Genbank format)

pDJ147.nucl: pDJ146 sequence (macVector format)

pDJ147.gb: pDJ146 sequence (Genbank format).

Click here for additional data file.

The authors thank the Bloomington Stock Center (National Institutes of Health P40OD018537), John Abrams, Scott Pletcher and John Tower for fly strains, and Liqun Luo and Jerry Yin for the pMS12, pJY2000 and pTub-GAL80 plasmids.

Additional Information and Declarations

Competing Interests

Author Contributions

The authors declare there are no competing interests.

Taylor Barwell performed the experiments, analyzed the data, wrote the paper, prepared figures and/or tables, reviewed drafts of the paper.

Brian DeVeale performed the experiments, contributed reagents/materials/analysis tools.

Luc Poirier performed the experiments, analyzed the data, contributed reagents/materials/analysis tools.

Jie Zheng and Frederique Seroude performed the experiments.

Laurent Seroude conceived and designed the experiments, performed the experiments, analyzed the data, contributed reagents/materials/analysis tools, wrote the paper, prepared figures and/or tables, reviewed drafts of the paper.

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
