# Peer review of "Regulating the UAS/GAL4 system in adult Drosophila with Tet-off GAL80 transgenes"

_PeerJ, doi:10.7717/peerj.4167_

## Round 0.1 · original submission · Major Revisions

Your manuscript has been read by two reviewers and there are several areas that must be addressed before it is suitable for publication:

1. The manuscript should be significantly edited and re-written to improve readability.
2. Please indicate specific P-values for all statistical tests
3. Please provide information on tetracycline treatment and on minimum effective dosage lifespan or behavior.
4. Provide information about the efficiency of the CMV1-TetO promoter in pDJ146/147
5. Rearrange some supplementary and result figures for better flow.
6. Please address all reviewers comments

·

Basic reporting

This is a good idea that introduces a new method for the temporal regulation of GAL4/UAS based transgene expression in Drosophila. The potential for this method is enormous in my opinion. I believe that this new approach to temporal transgene control will be of great utility to many fly researchers, but still need to present more “specific” GAL4 lines to proof this idea. However, the paper should be significantly edited; really they should re-write it entirely.

Experimental design

As describes, the results of UAS-grim lethality test reveals that it’s unstable lethality ratio displayed not only in each independent experiment but also under copy number of DJ146 or DJ147 (tubulin 1alfa and actin 5C promoter). The applicability of lethality caused from grim gene been concerned. Improving grim function or change lethal gene in here are suggested.

The author notices that insufficient Gal80 maybe cause by tTA uniformly distributed, molecule number of Gal80 (driver copy number), not overall overlap between Gal80 and grim expression region which under driver controlling. But it is interesting about that grim cannot blocked effectively by Gal80 even in the same tissue or same cells although Gal80 expression level is more far than the expression level of grim. Would you refer possible reasons to explain it?

I'm sure that the field would like to know how long it would take to turn expression off again by removing the drug from the food.

Validity of the findings

Does the "tet-GAL80 inducible system” by rtA-tetracycline showed more efficacy precisely expression? I suggest that more specific GAL4-driver confirm is needed. Would be better supported these evidences if the GFP fluorescence results were quantified and if the specific criteria used for the comparisons were defined, when compared with the conventional TARGET system by tub-GAL80ts under permissive temperature? The stereotyped nature of the olfactory lobes and mushroom bodies should make objective quantification of reporter in the relevant glomeruli/neuropils feasible for these assays (Kuo et al., PLoS One. 2012; 7(12):e50855). But even if the contention that TARGET system gave higher expression levels would be a most welcome addition to this study.

The authors should report specific P-values for all statistical tests, either in the results section or the figure legends.

Additional comments

The writing is confusing and the organization of the data is not very good either. I suggest that they fully re-write the paper starting with a results section that describes the reagents carefully, which does not need to be long but clear. The set of data is coherent, but too dense. The message they want to convey will be clearer after a serious trim showing a synthesis of the principal results. This ms should be rewrite or significantly edited.

Reviewer 2 ·

Basic reporting

No comment

Experimental design

No comment

Validity of the findings

No comment

Additional comments

In this manuscript, the authors describe a new approach to regulate gene expression in Drosophila using a tet-off Gal80 system. It complements the currently available tools such as Gene-switch or temperature sensitive Gal80 to control the temporal activity of Gal4. They provide detailed analyses on the new system in the result and supplementary sections. The tet-off Gal80 system requires multiple transgene (>2 copies) to obtain high level of Gal4 inhibition, which makes the system not very practical in its current state. However, it provides a proof of concept and a foundation for the research community to improve and optimize the system. There are some general concerns and suggestions to the authors that they may find useful.

General concerns:
1) Does the chronic tetracycline treatment have any detrimental effect? Do the authors have any information about the dosage effect on lifespan or behavior (e.g. minimum effective dosage)?

2) Do the author have any information about the efficiency of the CMV1-TetO promoter in pDJ146/147? The reported low efficiency in suppressing Gal4 using these two constructs comes a little bit surprising.

3) The authors may want to consider rearranging some of the supplementary and result figures. In fact, some of the supplementary results are more informative (e.g. SFig 13, 14 – demonstrate the effect of inducer; STable 15 – effect on various Gal4 lines) than result figures (e.g. Fig 2 - not very informative unless you also show the expression pattern with pDJ146/147, with/without inducer).


Specific comments:
1) The layout of Figure 1A is a little confusing. The authors could simplify the figure by showing progenies on one side (tet on) and the other side (tet off). “Hexagon” in the figure text should be “pentagon”.

2) It is very helpful for the authors to provide figures to every step in the construction of the transgenes. The authors could consider simplifying them by removing some of the irrelevant restriction sites but provide the final, full sequence information of the constructs pDJ146/147. The author may also want to consider using Gibson assembly to generate complex transgene constructs next time.

3) It is quite surprise to see pDJ147, with actin5c promoter, has no expression in the CNS (fig 6)? Could it be a positional effect specific to this transgenic line?

4) In figure 8, does the flies have UAS-grim? If not, it should be removed from the figure text?

5) In supplementary tables (e.g. STable 9,10), it would be better to show the actual measurement and highlight the one in bold to show the significance. Showing the P-value only without the actual number is not very informative.

6) The background of Figure 6 is too dark in some panels. It would be helpful if the authors can fix it.

---

## Round 0.2 · accepted · Accept

Revised manuscript is acceptable in the present form.

·

Basic reporting

No comment

Experimental design

No comment

Validity of the findings

No comment

Additional comments

This is a brilliant study that introduces a new method for the temporal regulation of GAL4/UAS based transgene expression in Drosophila. The potential for this method is enormous in my opinion. Authors replied all the points that I asked and showed reasonable results. I think this manuscript is ready to be published in PeerJ. I believe that this new approach to temporal transgene control will be of great utility to many fly researchers. It is a significant and welcome contribution to the field.